Disorder affects judgements about a neighbourhood: police presence does not

Hill Jessica 1 j.hill@nscr.nl
Pollet Thomas V. 2
Nettle Daniel 3
1 The Netherlands Institute for the Study of Crime and Law Enforcement (NSCR) , Amsterdam , The Netherlands
2 Department of Social and Organizational Psychology, VU University, Amsterdam , The Netherlands
3 Institute of Neuroscience, Newcastle University, Newcastle , UK
Abdullah Jafri
Electronic publication date: 2014 Mar 4
Publication date: 2014
Volume: 2
Electronic Location ID: e287
Received 2013 Dec 3; Accepted 2014 Feb 2
Copyright: © 2014 Hill et al.
Copyright year: 2014
Copyright holder: Hill et al.
License: This is an open access article distributed under the terms of the Creative Commons Attribution License, which permits unrestricted use, distribution, and reproduction in any medium, provided the original author and source are credited.
License URL: https://creativecommons.org/licenses/by/3.0/

Keywords: Social capital, Police visibility, Disorder, Fear of crime

Funding: NWO Research Talent NWO Veni 451.10.032 Jessica M. Hill is supported by an NWO Research Talent grant. Thomas V. Pollet is supported by NWO Veni grant (451.10.032). The funders had no role in study design, data collection and analysis, decision to publish, or preparation of the manuscript.

==============================
Many police forces operate a policy of high visibility in disordered neighbourhoods with high crime. However, little is known about whether increased police presence influences people’s beliefs about a neighbourhood’s social environment or their fear of crime. Three experimental studies compared people’s perceptions of social capital and fear of crime in disordered and ordered neighbourhoods, either with a police presence or no police presence. In all studies, neighbourhood disorder lowered perceptions of social capital, resulting in a higher fear of crime. Police presence or absence had no significant effect. The pervasive effects of disorder above other environmental cues are discussed.

Neighbourhoods within a single city can vary greatly in nature, from the state of buildings and the upkeep of public spaces to the frequency of police patrols (Nettle, Colléony & Cockerill, 2011). When we encounter an unfamiliar neighbourhood using cues from the physical environment helps us make judgements about strangers we may encounter there. Being able to read signals indicating a potentially unsafe environment and therefore the need to exercise vigilance would be adaptive. The aim of the studies presented in this paper was to examine whether cues from the physical environment influence people when forming judgements about the social environment and consequently determine how fearful for their safety they should be. Specifically, would naïve observers’ perceptions of a community be affected differently by disorder as opposed to order in a neighbourhood, and by the presence or absence of the police? With the adoption by many police forces of high visibility hot-spot policing (Boyd, Geoghegan & Gibbs, 2011), investigating whether this tactic may affect perceptions of the social environment is clearly pertinent.

Humans evolved to read environments quickly, unconsciously and to store environmental information in order to distinguish between survival enhancing and survival threatening landscapes (Kaplan, 1992). Naïve observers consistently form accurate impressions of occupants from their personal environments (Gosling et al., 2002). Harris & Brown (1996) found naïve observers were able to use information about the appearance of houses to accurately judge residents’ commitment to an area. At a neighbourhood level O’Brien & Wilson (2011) found naïve observers were able to accurately judge the social quality of a neighbourhood from aspects of its physical appearance, a concept they label ‘community perception’. When making judgements about residents, participants responded to generalised cues of effort invested in a neighbourhood’s appearance, such as upkeep of lawns and prevalence of litter. More physically disordered neighbourhoods were correctly believed to have poorer social quality than ordered neighbourhoods, with participants deeming residents less trustworthy partners in an economic game than residents of ordered neighbourhoods.

That neglected neighbourhoods are judged by residents and passers-by alike to be of inferior social quality corresponds with much work in the sociological and criminological fields. Various theories recognise the reciprocal relationships between neighbourhood disorder, a poor social environment, high crime rates and fear of crime (e.g., disorder theory: Wilson & Kelling, 1982; incivilities thesis: Hinkle & Weisburd, 2008; social disorganization theory: Sampson & Groves, 1989). The contagious nature of disorder is key to Wilson and Kelling’s (1982) disorder theory (also known as broken windows theory), whereby low-level disorder, such as a broken window, spreads and escalates. Social and physical disorder in a neighbourhood indicates to observers a community which does not behave appropriately and cannot maintain or enforce social norms. This contagiousness of disorder has been demonstrated empirically via a series of experiments by Keizer, Lindenberg & Steg (2008). In contrast, neighbourhoods where community networks are strong and norms are adhered to, have lower crime rates and residents who fear crime less (Sampson, Raudenbush & Earls, 1997). Greater feelings of personal safety are fostered in a cohesive and stable community (Taylor & Covington, 1993), where social ties are stronger.

Community social ties contribute to social capital (Putnam & Feldstein, 2003). Social capital is generally used to refer to social networks, norms of reciprocity, mutual assistance and trustworthiness within a community (Putnam & Feldstein, 2003). Residents in neighbourhoods high in social capital experience less fear of crime and feel safer, as residents trust their neighbours and believe them willing to implement social control (Gibson et al., 2002). Aspects of the physical environment which indicate low social capital would thus signal that caution may be required when interacting with residents, as this is a community of low trust where social norms are not always followed. In the current studies we therefore predicted that participants would perceive social capital within a community to be lower, and would be more fearful of crime, when presented with a disordered than an ordered neighbourhood. Furthermore, we predicted that perceived social capital would mediate the relationship between neighbourhood disorder and fear of crime.

Whilst the negative effects of disorder on people’s perceptions have been widely researched, other aspects of the environment typical of disordered neighbourhoods, such as high police presence, have not been so extensively examined. Several studies have looked at whether increasing police presence within a neighbourhood affects residents’ perceptions. Some reported a positive effect of police presence on community cohesion and trust (e.g., Bennett, 1991; Ferguson & Mindel, 2006; Scott, 2002), as well as a reduction in fear of crime (e.g., Zhao, Scheider & Thurman, 2002). High police visibility has also been linked to an increase in confidence in the police (Sindall & Sturgis, 2013). However, Weisburd and colleagues (2011) found that increased police presence in crime ‘hot spots’ did not have any effect on the collective efficacy or fear of crime of residents. They concluded that people did not really notice the police in their daily lives. Similarly, Mason (2009) reported that introducing neighbourhood policing had no positive effects on residents’ confidence in the police, again possibly due to a lack of awareness of police presence (Brunton-Smith, Sutherland & Jackson, 2013). In an experimental study carried out in the Netherlands, Van de Veer and colleagues (2012) found an interaction effect between police presence and background environment, with participants’ fear of crime increasing when police were present in a ‘safe’ environment, and decreasing when police were present in an ‘unsafe’ environment. In contrast, Kochel (2011) discusses how the presence of police in crime hot-spots can have a negative effect. Police patrol cars, as opposed to foot patrols, have been shown to increase fear of crime (Salmi, Grönroos & Keskinen, 2004). Police presence, particularly when ‘buffered’ from residents in a vehicle, may signal to passers-by a community which does not follow social norms, is not trustworthy, and therefore needs policing. This indication of low social capital would thus lead to greater fear of crime in ‘unsafe’ areas with a highly visible police presence.

Due to the contradictory findings of the previous research outlined we present three competing predictions for the effect of police presence on perceptions of a neighbourhood and its community. Based on Weisburd et al. (2011) and Mason’s (2009) findings we might predict that the presence of the police would not signal anything over and above other cues from the physical environment, and therefore not affect perceptions of social capital or fear of crime. Alternatively, our prediction based on Van de Veer et al.’s findings would be that people would fear crime less in a disordered neighbourhood when police were present, but more in an ordered neighbourhood when police were present. On the other hand, if the presence of police is perceived as an indication of lower social capital within the neighbourhood, and social capital acts as a mediator on fear of crime, we predict their presence would lead to an increase in fear of crime across neighbourhoods (see Table 1) (2012).

Table 1 Effects of police presence in disordered and ordered neighbourhoods.

Competing predictions of the effects of police presence in disordered and ordered neighbourhoods on fear of crime.

	Disordered	Ordered	
	Police absent	Police present	Police absent	Police present	
Prediction 1	High fear of crime	No change	Low fear of crime	No change	
Prediction 2	High fear of crime	Decrease in fear of crime	Low fear of crime	Increase in fear of crime	
Prediction 3	High fear of crime	Increase in fear of crime	Low fear of crime	Increase in fear of crime	

We carried out three studies comparing disordered with ordered neighbourhood environments, either with or without the presence of police, in order to test our predictions regarding the effect of neighbourhood order, as well the three competing predictions regarding police presence. The first study used written descriptions of neighbourhoods; the second and third relied on visual cues. In all studies participants’ perceptions of the social capital of residents, as well as their perceived fear of crime, were measured.

Study 1

Study 1, an internet study with participants resident in the USA, used written descriptions of neighbourhoods, one disordered and one ordered, which included either sentences describing the presence of police patrol cars, or no references to the police. Measures of perceived social capital of residents (adapted from Sampson, Raudenbush & Earls, 1997; Sampson & Raudenbush, 1999) and fear of crime, operationalized as feelings of safety, were utilized.

Method

Participants

Participants were recruited via the crowd sourcing website, Crowdflower. In total 251 participants residing in the USA completed the survey. They each received $0.20 for participating. Of these 103 participants were excluded from the analyses due to spending less than 15 s or more than two minutes reading the description of neighbourhood. These cut-offs were used as we felt that participants could not realistically have read the vignettes in less than 15 s, and that if they had spent longer than two minutes ‘reading’ the vignettes it was possible they were not concentrating solely on the study. Ten participants were excluded due to not completing all the dependent measures. Of the remaining 138 participants 51% were female. The majority (38%) were aged between 20 and 29 years.

Procedure

Before beginning participants were told solely that they were taking part in a study about neighbourhood deprivation. After having given consent they proceeded to the study. Once completed participants were informed of the true nature of the study, i.e., that we were comparing the effect of affluent and deprived neighbourhoods, with either high police visibility or no police visibility, on social capital and fear of crime. Participants were given contact details should they desire further information.

Materials

Participants read a short vignette describing either a disordered or an ordered neighbourhood. The ordered neighbourhood vignette describes a high street, on which there is, amongst others, a delicatessen, a bank and a hotel, as well as a well-kept residential street. The disordered neighbourhood vignette describes a high street on which there is, among others, a liquor store, a tattoo parlour and some boarded-up store fronts, as well as a rundown residential street. Half of the vignettes presented in each neighbourhood description included three references to police presence, e.g., as you walk along the high street a police patrol car drives past. The same police references were used for the disordered and ordered neighbourhoods (See Appendix S1 for vignettes in full).

Measures

After reading one of the four possible vignettes, participants rated aspects of the social environment of the neighbourhood to measure their perceptions of residents’ social capital. One item measured how much participants felt residents could be trusted, responding on a slider scale of 0–100. Two items measured perceptions of informal social control: ‘If there were a fight in this neighbourhood residents would interfere’ and ‘if children were skipping school and hanging out on a street corner, residents would not take any action’ (reversed). Three items measured the perceived social cohesion: ‘People in this neighbourhood generally do not get along with each other’ (reversed), ‘people in this neighbourhood share the same values’, and ‘people around here are willing to help their neighbours’. Social control and cohesion were measured using a 5 point Likert scale (1 = strongly disagree, 5 = strongly agree). Fear of crime was measured by asking how safe participants would feel walking alone through the neighbourhood. Responses were on a 5 point Likert scale (1 = not at all safe, 5 = very safe).

The study was approved by the Faculty of Medical Sciences ethics committee, Newcastle University.

Results

The six social capital-related measures (trust, 2 measures of social control and 3 measures of social cohesion) were all significantly positively correlated with one another (rs.17 to .66; all p < .05). They were standardised and summed to provide an overall measure of social capital (SD = 4.3), with high reliability (Cronbach’s α = .81).

Two-way ANOVAs were carried out with neighbourhood order and police presence both between-subjects factors. Neighbourhood order had a significant effect on participants’ perceptions of social capital, with lower social capital reported by participants in the disordered neighbourhood conditions (M = −.58, SD = .68), than by those in the ordered neighbourhood conditions (M = .65, SD = .6), F(3, 134) = 128.51, p < .001, η2 = .49. Police presence had no significant effect on participants’ perceptions of social capital (police M = −.02, SD = .86, no police M = .02, SD = .92; F[3, 134] = 0.06, p = .8, η2 < .001). There was no significant interaction between neighbourhood order and police presence on perceptions of social capital (F[3, 134] = 0.8, p = .3, η2 = .006) (see Fig. 1).

Figure 1 Perceived social capital and fear of crime, Study 1.

(A) Mean perceptions of social capital by neighbourhood order and police presence, Study 1. Error bars represent 95% CI. (B) Mean fear of crime by neighbourhood order and police presence, Study 1. Error bars represent 95% CI.

Neighbourhood order had a significant effect on participants’ feelings of safety, with those in the disordered neighbourhood conditions feeling less safe (M = −.67, SD = .79), than those in the ordered neighbourhood conditions (M = .75, SD = .59), F(3, 134) = 139.09, p < .001, η2 = .51. Police presence had no significant effect on participants’ feelings of safety (police M = −.01, SD = 1.03, no police M = .01, SD = .96; F[3, 134] = 0.25, p = .6, η2 = .002). There was no significant interaction between neighbourhood order and police presence on feelings of safety (F[3, 134] = 0.007, p = .9, η2 < .001) (see Fig. 1).

Mediation analysis was carried out to test whether perceived social capital statistically mediated the effect of neighbourhood order on participants’ feelings of safety. Neighbourhood order significantly affected feelings of safety and perceptions of social capital (see above). The direct effect of perceptions of social capital on feelings of safety whilst keeping constant neighbourhood order was significant, B = 0.99, t(135) = 10.64, p < .001, η2 = .47. A Sobel test indicated mediation was significant, Sobel = 7.77, p < .001. The direct effect of neighbourhood disorder on feelings of safety whilst keeping constant perceptions of social capital remained significant, although weakened (B = 0.67, t[135] = 4.09, p < .001, η2 = .18), and so mediation was partial.

Discussion

Participants who read a vignette describing a disordered neighbourhood perceived residents’ social capital to be lower and felt less safe, i.e., had higher fear of crime, than participants who read a vignette describing an ordered neighbourhood. Perceived social capital partially mediated the relationship between neighbourhood and feelings of safety. Whether there was a reference to the presence of police in the vignette or not had no significant effect on perceptions of social capital or feelings of safety, for participants reading either the disordered or the ordered neighbourhood description. These results indicate that disorder within a neighbourhood signals to naïve observers a social environment of poor quality, and that this is therefore an unsafe place where crime is to be feared. However, when making judgements about residents of an unfamiliar neighbourhood people do not appear to be using police presence as a cue to the nature of the social environment.

One limitation of this study was the use of written descriptions. As we usually assess neighbourhoods visually these lack ecological validity. We therefore conducted another study using visual stimuli.

Study 2

Study 2 used a series of photographs as stimulus material. The photographs used were of two neighbourhoods in the city of Newcastle-upon-Tyne, UK. These neighbourhoods have been extensively studied for comparing outcomes and behaviour in a socioeconomically deprived and an affluent neighbourhood (Nettle, Colléony & Cockerill, 2011; Nettle, Coyne & Colléony, 2012; Nettle, 2011; Nettle, 2012). The physical environments in these neighbourhoods, in terms of maintenance of public spaces, businesses and housing upkeep, are highly contrasting (see Fig. 2). The same outcome measures were employed as in Study 1.

Figure 2 Example photos, Study 2 and 3.

(A) Disordered neighbourhood without police presence; (B) Ordered neighbourhood without police presence; (C) Disordered neighbourhood with police presence; (D) Ordered neighbourhood with police presence.

Method

Participants

In total 60 participants (77% female) residing in the UK completed the study in a laboratory setting. Participants’ ages ranged from 19 to 63 years (M = 33.3, SD = 12.6). Participants were recruited from the university participant pool and were all familiar with psychological laboratory experiments. Participants received £5 for participation. No participants were resident in the neighbourhoods featured in the slideshows; the majority were resident in the metropolitan area of Tyne and Wear.

Procedure

Materials

Four different slideshows were presented on a computer screen; two showing 40 photographs of a disordered neighbourhood, two showing 40 photographs of an ordered neighbourhood. In the police-present conditions slideshows included 10 photographs of police cars patrolling the neighbourhood. The police-absent conditions slideshows included photographs of the same scenes photographed once the police cars had moved on. Each photograph was displayed for ten seconds, creating slideshows of 6 min 40 s.

Measures

After viewing a slideshow, participants rated aspects of the social environment of the neighbourhood they had just seen. The six items from Study 1 were used to measure perceived social capital of residents, all measured using a 5 point Likert scale. Fear of crime was also measured as in Study 1 by asking participants how safe they would feel in the neighbourhood.

Procedure

A mixed study design was employed, with neighbourhood order a within-subjects factor and police presence a between-subjects factor. Participants were informed that they were taking part in a study on neighbourhood deprivation. Once consent had been given participants watched the first slideshow. As they watched they were asked to count the number of cars with visible number plates, a check to ensure focus on the photographs. They then responded to the perceived social capital and fear of crime measures for the neighbourhood just viewed. This was followed by a second slideshow, with the same counting task, and the perceived social capital and fear of crime measures for this neighbourhood. Half the participants viewed a disordered neighbourhood slideshow first, half an ordered. Demographic information was collected after viewing the slideshows. Finally, participants were fully debriefed, thanked for their participation and received payment.

The study was approved by the Faculty of Medical Sciences ethics committee, Newcastle University.

Results

The six measures of trust, social control (two measures) and social cohesion (three measures) were summed to provide a measure of social capital (disordered conditions M = 291.3, SD = 70.7, ordered conditions M = 387.2, SD = 71.1). Reliability was moderately high in the disordered conditions (Cronbach’s α = .7) and ordered conditions (Cronbach’s α = .7).

Repeated measures ANOVAs were carried out with neighbourhood order as within-subjects factor and police presence as between-subjects factor. Neighbourhood order had a significant effect on participants’ perceptions of social capital, with lower social capital reported by participants in the disordered neighbourhood conditions (M = 48.1, SD = 12.02), than by those in the ordered neighbourhood conditions (M = 65.1, SD = 11.82), F(1, 46) = 14.16, p < .001, η2 = .24. Police presence had no significant main effect on participants’ perceptions of social capital (F[1, 46] = 0.74, p = .4, η2 = .02). There was no significant interaction between neighbourhood order and police presence on perceptions of social capital (F[1, 46] = 0.005, p = .9, η2 < .001) (see Fig. 3). Participant gender had no significant effect on perceptions of social capital (F[1, 46] = 0.87, p = .4, η2 = .018). There was a significant interaction between participant gender and neighbourhood order on perceptions of social capital F(1, 46) = 7.78, p < .008, η2 = .15. In the ordered neighbourhood conditions females perceived social capital as higher (M = 66.7, SD = 12.4) than males (M = 60, SD = 8.2); in the disordered neighbourhood females perceived social capital as lower (M = 45.5, SD = 12.2) than males (M = 56.3, SD = 6.8). There was no significant interaction between participants gender and police presence on perceptions of social capital (p = .6).

Figure 3 Perceived social capital and fear of crime, Study 2.

(A) Mean perceptions of social capital by neighbourhood order and police presence, Study 2. Error bars represent 95% CI. (B) Mean fear of crime by neighbourhood order and police presence, Study 2. Error bars represent 95% CI.

Neighbourhood order had a significant effect on participants’ feelings of safety, with those in the disordered neighbourhood conditions feeling less safe (M = 34.52, SD = 22.17), than those in the ordered neighbourhood conditions (M = 80.38, SD = 13.27), F(1, 46) = 87.84, p < .001, η2 = .66. Police presence had no significant main effect on participants’ feelings of safety (F[1, 46] = 0.46, p = .5, η2 = .001). There was no significant interaction between neighbourhood order and police presence on feelings of safety (F[1, 46] = 0.1, p = .9, η2 = .001) (see Fig. 3). Participant gender had a significant effect on feelings of safety, with females feeling less safe than males, F(1, 46) = 9.41, p = .004, η2 = .17. Participant gender did not significantly interact with either neighbourhood order or police presence on feelings of safety (p > .2).

Mediation analysis, following the method outlined for within-subject designs by Judd and colleagues (2001) was carried out to test whether perceived social capital mediated the effect of neighbourhood order on participants’ feelings of safety. Perceived social capital can be said to mediate feelings of safety if two conditions are met. First, the difference between neighbourhoods in perceived social capital must be in the same direction as feelings of safety. Perceived social capital was significantly related to feelings of safety for disordered neighbourhood, B = 0.96, t(53) = 4.05, p < .001, η2 = .49, and the ordered neighbourhood, B = 0.73, t(53) = 5.58, p < .001, η2 = .61, with higher social capital indicating higher feelings of safety and vice versa. Second, the difference in perceived social capital must significantly predict difference in feelings of safety. This it does, B = 0.95, t(53) = 6.09, p < .001, η2 = .65. Mediation was partial, as the residual difference in feelings of safety between neighbourhoods remained significant over and above neighbourhood difference in perceived social capital (B = 44.35, t[53] = 11.33, p < .001).

Discussion

The results of Study 2 mirrored those of Study 1, with participants perceiving residents of the disordered neighbourhood as having less social capital than residents of the ordered neighbourhood. Participants reported feeling less safe, i.e., a higher fear of crime, when viewing the disordered rather than the ordered neighbourhood. This effect was partially mediated by perceptions of social capital. Police presence had no effect on participants’ perceptions of social capital or feelings of safety for either the disordered or the ordered neighbourhood. The results of this study again indicate that whilst naïve observers use aspects of the physical environment to inform their judgements about the social environment of a neighbourhood, consequently affecting their fear of crime, police presence is not a cue that affects these judgements.

Gender had a significant effect on fear of crime, with females feeling less safe across all conditions. That females have higher fear of crime is unsurprising (Jackson, 2009). Interestingly though, there was a significant interaction between gender and neighbourhood order on participants’ perceptions of social capital. However, due to the small number of male participants in the study (n = 14) caution when interpreting these results is required. Further investigation of gender effects using a larger male sample would therefore be beneficial. Furthermore, the study by Van de Veer and colleagues (2012) found that the presence of police had a stronger effect on male feelings of safety than female, possibly, they concluded, because men are the cause of police presence more frequently than women, either as perpetrator or victim.

Another relevant individual difference variable, which may influence people’s perceptions of police presence, is childhood environment. A child who has grown up in a safe environment having little contact with crime or experience of disorder, might, as an adult, react differently to the presence of police than a person who grew up in an unsafe environment, where disorder and crime were more common. We carried out a further study to address these two issues.

Study 3

Study 3 was conducted online, using British participants. Procedurally this study was similar to Study 2, but the key differences were that participants viewed only one slideshow, the slideshows were shorter in length, and survey measures of childhood SES were taken. Childhood SES was collected as a means of determining childhood environment, with the assumption that a higher childhood SES reflected a safer childhood environment and vice versa. In addition a larger sample was collected to ensure more male participants.

Method

Participants

Participants were recruited via the crowd sourcing website, Crowdflower. In total 169 participants residing in the UK responded to the survey. They each received $0.50 for participating. Fifty participants were excluded from the analyses due to unsatisfactory completion of the measures. Of the remaining 119 participants 79% were male. They ranged in age from 17 to 55 years old (M = 32.2, SD = 11.9).

Procedure

After providing consent participants viewed a slideshow. The photos from Study 2 were reused, with each displayed for 5 rather than 10 s, resulting in slideshows of 3 min 20 s. Participants were asked to either count red cars or police cars (depending on the slideshow viewed), as a check in order to ensure focus on the photographs. Social capital and fear of crime were measured as in Studies 1 and 2. Childhood SES was measured by asking participants to respond on a Likert scale of 1–7 the extent to which they agreed with the following three statements: My family usually had enough money for things when I was growing up; I grew up in a relatively wealthy neighbourhood; I felt relatively wealthy compared to other kids in my school (Griskevicius et al., 2011). Once demographic details had been collected participants were fully debriefed, thanked for their participation, given space to comment on the task, and provided with researcher contact details should they desire further information.

The study was approved by the Faculty of Medical Sciences ethics committee, Newcastle University.

Results

The six measures of trust, social control (two measures) and social cohesion (three measures) were all significantly positively correlated with one another (rs 0.24–0.57; all p < .05). They were summed to provide an overall measure of social capital (M = 330.8, SD = 83.2), with moderate reliability (Cronbach’s α = .65). The three measures of childhood SES were all significantly positively correlated with one another (rs 0.6–0.75). They were summed to provide an overall measure of childhood SES (M = 11.9, SD = 4.5), with high reliability (Cronbach’s α = .82).

Two-way ANOVAs were carried out with neighbourhood order and police presence both between-subjects factors. Neighbourhood order had a significant effect on participants’ perceptions of social capital, with lower perceived social capital reported by participants in the disordered neighbourhood conditions (M = 49.3, SD = 12.8), than by those in the ordered neighbourhood conditions (M = 62, SD = 12.1), F(5, 111) = 29.68, p < .001, η2 = .21. Police presence had no significant effect on participants’ perceptions of social capital (police M = 54, SD = 14.1, no police M = 55.8, SD = 13.9; F[5, 111] = 0.66, p = .4, η2 = .006). There was no significant interaction between neighbourhood order and police presence on perceptions of social capital (F[5, 111] = 0.001, p = .97, η2 < .001) (see Fig. 4).

Figure 4 Perceived social capital and fear of crime, Study 3.

(A) Mean perceptions of social capital by neighbourhood order and police presence, Study 3. Error bars represent 95% CI. (B) Mean fear of crime by neighbourhood order and police presence, Study 3. Error bars represent 95% CI.

Participant gender had no significant effect on perceptions of social capital (male M = 55.7, SD = 13.4, female M = 51.9, SD = 15.9; F[7, 109] = 0.41, p = .5, η2 = .004). There was a significant interaction between gender and neighbourhood order on perceptions of social capital, F(7, 109) = 12.6, p < .001, η2 = .19. Social capital was perceived similarly by females (M = 61.7, SD = 9.2) and males (M = 62.1, SD = 12.8) in the ordered neighbourhood conditions. In the disordered neighbourhood conditions, females perceived social capital to be lower (M = 45, SD = 16.1) than males perceived social capital to be (M = 50.6, SD = 11.6). There was no significant interaction between gender and police presence on perceptions of social capital (F[7, 109] = 0.24, p = .8, η2 = .004). There was no significant interaction between gender, neighbourhood order and police presence (F[7, 109] = 0.88, p = .4, η2 = .016).

Childhood SES had no significant effect on perceptions of social capital (F[4, 112] = 1.62, p = .2, η2 = .014). There was no significant interaction between childhood SES and neighbourhood order on perceptions of social capital (F[4, 112] = 0.03, p = .9, η2 < .001). There was no significant interaction between childhood SES and police presence on perceptions of social capital (F[4, 112] = 0.1, p = .9, η2 < .001). There was no significant interaction between childhood SES, neighbourhood order and police presence on perceptions of social capital (F[4, 112] = 0.54, p = .5, η2 = .005).

Neighbourhood order had a significant effect on participants’ feelings of safety, with those in the disordered neighbourhood condition feeling less safe (M = 38.9, SD = 25.3), than those in the ordered neighbourhood condition (M = 73, SD = 20.5), F(5, 111) = 63.08, p < .001, η2 = .36. Police presence had no significant effect on participants’ feelings of safety (police M = 49.9, SD = 29.4, no police M = 57.6, SD = 28.1; F[5, 111] = 2.74, p = .1, η2 = .02). There was no significant interaction between neighbourhood order and police presence on feelings of safety (F[5, 111] = 0.34, p = .6, η2 = .003) (see Fig. 4).

Participant gender had no significant effect on feelings of safety (male M = 56.6, SD = 27.8, female M = 44.2, SD = 31.1; F[7, 109] = 1.56, p = .2, η2 = .015). There was a significant interaction between gender and neighbourhood order on participants’ feelings of safety, F(7, 109) = 27.43, p < .001, η2 = .34. Females (M = 70.8, SD = 17.1) and males (M = 73.6, SD = 21.4) reported similar feelings of safety in the ordered neighbourhood conditions. In the disordered neighbourhood conditions, females’ feelings of safety were lower (M = 25.2, SD = 23.9) than males’ feelings of safety (M = 42.6, SD = 24.6). There was no significant interaction between gender and police presence on feelings of safety (F[7, 109] = 1.23, p = .3, η2 = .022). There was no significant interaction between gender, neighbourhood order and police presence on feelings of safety (F[7, 109] = 1.41, p = .3, η2 = .025).

Childhood SES had no significant effect on participants’ feelings of safety (F[4, 112] = 0.12, p = .7, η2 = .001). There was no significant interaction between childhood SES and neighbourhood order on participants’ feelings of safety (F[4, 112] = 0.23, p = .6, η2 = .002). There was no significant interaction between gender and police presence on participants’ feelings of safety (F[4, 112] = 1.66, p = .2, η2 = .015). There was no significant interaction between gender, neighbourhood order and police presence participants’ feelings of safety (F[4, 112] = 1.34, p = .3, η2 = .012).

Mediation analysis was carried out to test whether perceived social capital mediated the effect of neighbourhood order on participants’ feelings of safety. Neighbourhood order significantly affected feelings of safety and perceived social capital (see above). The direct effect of perceived social capital on feelings of safety whilst keeping constant neighbourhood order was significant, B = 1.11, t(114) = 785, p < .001, η2 = .48. A Sobel test revealed mediation was significant, Sobel = 4.48, p < .001. The direct effect of neighbourhood disorder on feelings of safety whilst keeping constant social capital remained significant, although weakened (B = 20.19, t[114] = 5.13, p < .001, η2 = .31), indicating that mediation was partial.

Discussion

The results of Study 3 show that whilst disorder had an effect on peoples’ perceptions of the social environment in a neighbourhood, and consequently their feelings of safety i.e., fear of crime, the presence of police had no effect on these, either in disordered or ordered neighbourhoods. Participant gender and neighbourhood order had a significant interaction effect on both perceptions of social capital and feelings of safety, with females in the disordered neighbourhood conditions perceiving social capital to be lower and reporting feeling less safe than males in the disordered conditions. However, the presence of the police did not affect male or female participants’ perceptions of the social environment or their feelings of safety differently. Childhood SES did not have any significant effect on participants’ perceptions of social capital or their feelings of safety.

General Discussion

The findings of all three studies demonstrate that when people were presented with a disorderly neighbourhood they judged the social environment to be of poorer quality, i.e., residents had lower social capital, and were consequently more fearful for their safety, than when they were presented with an ordered neighbourhood. This effect of the disorder on judgements was stronger for females than males. The main effects of neighbourhood disorder come as no surprise considering the wealth of theory and research on the negative influence disorder can have on a community and on outsiders’ perceptions of a community, as outlined in the introduction. The aim of the current studies was not to test the accuracy of people’s judgements of residents’ social capital, and whether, as O’Brien & Wilson (2011) found, ‘community perception’ as an adaptive mechanism was at work. Nevertheless, previous research carried out in the neighbourhoods in Newcastle-upon-Tyne, photographs of which were used as stimulus material in Studies 2 and 3, suggests participants were making accurate judgements about the social environment (Nettle, Colléony & Cockerill, 2011). Nettle and colleagues found residents of the disordered neighbourhood reported lower social capital than residents of the ordered neighbourhood. The conclusion that community perception was demonstrated, that people used cues from the physical environment to accurately interpret the quality of the social environment, can thus be cautiously drawn.

In the introduction we presented three competing predictions for the effect of police presence. All three studies clearly demonstrated that the presence of the police had no significant, measurable effect on people’s judgements about social capital within the neighbourhoods or their fear of crime. This held true across both neighbourhoods, for men and women, regardless of whether they had a poor or affluent childhood. Our finding that police presence had no significant effect on naïve observers’ perceptions over and above the effect of disorder or order within a neighbourhood is consistent with findings from Mason’s (2009) report on neighbourhood policing, as well as Weisburd and colleagues (2011), who found residents’ social capital, and fear of crime, remained unchanged when police presence was increased. They concluded that unless people are directly impacted by the police, their presence goes unnoticed and that, at least in the short term, an increase in police numbers on the streets does not affect a community, either positively or negatively. In our case, it is possible that disorder in a neighbourhood was sending a strong signal to passers-by that an area is of poor social quality, and vice versa for neighbourhoods with visible signs of affluence. Such that if disorder triggers preconceived ideas about the nature of a community, ‘extra’ information from the environment may then be overlooked. Sampson & Raudenbush (2004) found that the social structure of a neighbourhood, in their study predominantly determined by race, was a better predictor of people’s perceptions of disorder than actual observed disorder. Franzini and colleagues (2008) similarly found that neighbourhood poverty affected people’s perceptions of disorder, but also that perceived disorder was to an extent in the eye of the beholder: The higher educated and more residentially mobile perceived less disorder. These studies indicate that cultural stereotypes influence perceptions about a neighbourhood. If robust, but unconscious, heuristics are employed, such as ‘disorder in a neighbourhood means people cannot be trusted’, whether the police are present or not would therefore have little sway on judgements one way or the other.

It is, however, important to note that the studies carried out here investigated the effects of police in patrol cars. Previous studies where positive effects of police presence have been found generally examined police foot patrols (e.g., Salmi, Grönroos & Keskinen, 2004; Sindall & Sturgis, 2013). By patrolling on foot police become more approachable, removing a barrier between themselves and residents. This may consequently have an effect on observers’ impressions of a community. Future research is therefore required in order to determine whether the impressions naïve observers form of a community are influenced by police foot patrols.

A further limitation of this research is the relatively small sample sizes used in all three studies. Whilst we feel that the replication of results across the three studies demonstrates the strength of our findings, we acknowledge the limitation of the sample sizes we had. Another limitation of our studies is that we did not pre-test the vignettes or photos to ensure they were recognised as disordered or ordered. Whilst our results seem to indicate that they were perceived as such, pre-testing would have been desirable. Furthermore, whilst anecdotally we found participants noticed the police presence in the photographs used in Studies 2 and 3, we included no check for this. It is possible that participants did not notice the police cars, recognise them as such, or perceive that their presence indicated anything other than just passing through on a call. Further research is therefore required in order to test whether the police presence is understood to indicate patrolling the neighbourhood in question.

Furthermore we recognise that whilst our results indicate people’s initial perceptions of a neighbourhood are not affected by police presence, the long term effects of high police visibility on disordered neighbourhoods and their residents are likely to be different. If a hot spot policing approach is taken, i.e., increasing police presence in disordered neighbourhoods, and leads to a decrease in physical and social disorder, as indicated by past research (e.g., Braga, 2005), this would clearly in turn have a positive effect on people’s perceptions of a neighbourhood and its residents.

The role of the police is, not just to fight crime, but also to reduce fear of crime (Boyd, 2012). Police often face calls from politicians and the media, as well as from the public (Allen, 2004), to increase their visibility on the streets, in the belief that this will reduce crime and consequently fear of crime. Our research indicates, however, that police on the streets, at least in patrol cars, do not have an impact people’s fear of crime when in an unfamiliar area. What undoubtedly does influence people’s fear of crime for the worse is disorder, in part because people perceive residents of disordered neighbourhoods to have lower social capital. Tackling disorder, as argued Jackson and colleagues (2009) could be a more effective means to reducing fear of crime. In an era where public service budgets are becoming more limited, using empirical evidence to inform how we tackle societal problems such as fear of crime surely makes sense. If the police are using policies of high visibility to paint a picture of a safe neighbourhood to the outside world it seems they could be putting their resources to better use.

Supplemental Information

Appendix S1 Study 1 vignettes, for disordered and ordered neighbourhoods, with and without police presence

Click here for additional data file.

Additional Information and Declarations

Competing Interests

Author Contributions

Human Ethics

The authors declare no competing interests.

Jessica Hill conceived and designed the experiments, performed the experiments, analyzed the data, wrote the paper.

Thomas V. Pollet and Daniel Nettle conceived and designed the experiments, contributed reagents/materials/analysis tools, wrote the paper.

The following information was supplied relating to ethical approvals (i.e., approving body and any reference numbers):

Faculty of Medical Sciences Ethics Committee, Newcastle University, Approval number: 00616/2012

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
