# Peer review of "Disorder affects judgements about a neighbourhood: police presence does not"

_PeerJ, doi:10.7717/peerj.287_

## Round 0.1 · original submission · Minor Revisions

Merry Christmas and Happy Holidays.I hope these minor revisions can be done so as the manuscript can be rereviewed after the new year.

Reviewer 1 ·

Basic reporting

Overall, the authors do a good job of placing their research within the larger research field and discuss relevant prior literature. I feel that some theories such as Broken windows theory could be discussed in greater detail (and perhaps less emphasis could be placed on whether participants are accurate in their perceptions of neighbourhoods).

My main concern in the reporting in the introduction and the discussion is that a clear definition of the term social capital is missing. Given that it is such a central concept in the research I would expect a more detailed discussion of this concept. Also, especially in the discussion, social capital seems to be equated with more generally the quality or general perceptions of a neighbourhood. I think the authors could be more exact in their use of the term.

Some minor issues:
-In line 20 the wording is not entirely clear to me (I think the authors mean that order affects perceptions, not necessarily that order and disorder affect perceptions differently? Which seems to suggest that two different processes are going on.)
- In line 430 the word “also” appears to be missing after “ but”.

Experimental design

Overall, I think the methods could be explained in greater detail.

In study 1, I think the vignettes should also be shortly described in the material section, not only in the appendix.

It is not immediately clear from the measures section what the measure of social capital is. One example of the items of all of the measures would be nice.

In study 1 only an age range of participants is given, in study 2 and 3 a mean of age of participants is reported. I would keep the reporting consistent across studies (and report mean rather than range).

Were the vignettes in Study 1 and photo’s in Study 2 and 3 pretested to be sure these could indeed be classified as disordered and ordered?

What kind of instructions did participants get in the different studies? Was some kind of cover story used?

Why was the counting car number plates task included in Study 2 and 3? To me this seems to take away attention from perception of the neighbourhood and police or induce cognitive load rather than ensure a focus on the photographs.

Related to this, was it checked whether participants actually perceived the police/ perceived the car as being a police car in Study 2 and 3? To me the car/van is not immediately recognizable as police (especially if I was asked to focus on the number plate, but this could be related to the photograph being black and white/ not being native uk). In the discussion it is mentioned that police presence often goes unnoticed and that this is an explanation for null findings in field studies. Could this be going on in the present findings as well?

Validity of the findings

In the result sections, no mention is made of the type of statistical tests that were performed.

Measure of social capital is z transformed in Study 1 but not in the other two studies.

In the text the neighbourhood conditions are referred to as ordered vs disordered whereas in the figures they are referred to as deprived vs affluent.

Additional comments

Interesting set of studies, was interesting to read.

Reviewer 2 ·

Basic reporting

The article is well-written and easy to follow. There are no concerns with basic reporting.

Experimental design

The biggest concern here is whether this submission falls within the scope of the journal. As a criminologist, I find the work interesting and worthy of publication, but it does not seem to be applicable to the biological, medical, or health sciences and the authors have not made any explicit links between those findings and these areas. Because my own background is not in these areas, I cannot fully judge the applicability of this research to those areas, but this is one area of concern for whether this publication belongs in PeerJ.

The three experiments are all clearly described. The main concern with all three is that the sample sizes are fairly small and the authors may want to note that as one potential limitation. In Study 1 in particular, there seems to be a great deal of attrition as a result of excluding respondents. The authors may want to describe in a bit more detail why the cutoffs of 15 seconds and 2 minutes were used for determining when to remove a subject. I understand the desire to ensure that the subject actually read the vignette, but it's not as clear to me why anyone spending longer than 2 minutes on the page should be eliminated.

Validity of the findings

The findings are reported very clearly and based on the analyses, the conclusions drawn about the impact of police presence seem warranted (with the caveats noted below).

As the authors note, examining the impact of foot vs. car patrol would be useful in future research and the authors should be cautious in overstating their findings. In the pictures in particular, it seems that the police are simply driving down the block. This is certainly an indicator of police presence, but it doesn't tell respondents very much about how much time officers are actually spending in these places. It doesn't even demonstrate whether officers are actually stopping in these places or just passing through on their way to another call or location. While that doesn't invalidate the findings or conclusions here, it suggests that respondents may not always be equating the police with police presence in that particular place. Just seeing a police car driving down a street doesn't mean that the police are visiting that particular street.

One other area where some adjustment would be useful is the final conclusion that high visibility may not be a useful way to affect citizen perceptions of the police. Sometime to consider here would be how, in the long-term, high visibility policing would affect levels of disorder. Based on what we know from hot spots policing, the police can decrease levels of physical and social disorder in places when they increase their presence. And these decreased levels of disorder may then improve citizen perceptions of these places. This is of course a more long-term process than can be assessed in a single photograph, but it may be worth noting as a possible longer-term positive consequence of increasing presence.

Additional comments

No additional comments

---

## Round 0.2 · accepted · Accept

Dear Authors,

Thank you for your revised manuscript which has been accepted for publication.

·

Basic reporting

No comments

Experimental design

No comments

Validity of the findings

No comments

Additional comments

Thanks to the authors for their careful attention to my prior comments. I enjoyed reading the revised manuscript and have no further suggested revisions.